# *Lactobacillus johnsonii* L531 Ameliorates *Escherichia coli*-Induced Cell Damage via Inhibiting NLRP3 Inflammasome Activity and Promoting ATG5/ATG16L1-Mediated Autophagy in Porcine Mammary Epithelial Cells

**DOI:** 10.3390/vetsci7030112

**Published:** 2020-08-14

**Authors:** Yun-Jing Zou, Jia-Jia Xu, Xue Wang, Yao-Hong Zhu, Qiong Wu, Jiu-Feng Wang

**Affiliations:** 1Department of Veterinary Clinical Sciences, College of Veterinary Medicine, China Agricultural University, Beijing 100193, China; B20173050392@cau.edu.cn (Y.-J.Z.); jajaxu@126.com (J.-J.X.); IVYWANG0913@163.com (X.W.); zhu_yaohong@hotmail.com (Y.-H.Z.); 2College of Animal Science and Technology, Beijing University of Agriculture, Beijing 102206, China

**Keywords:** *Escherichia coli*, porcine mammary epithelial cell, inflammasome, autophagy, probiotic

## Abstract

*Escherichia coli* (*E. coli*), a main mastitis-causing pathogen in sows, leads to mammary tissue damage. Here, we explored the effects of *Lactobacillus johnsonii* L531 on attenuating *E. coli*-induced inflammatory damage in porcine mammary epithelial cells (PMECs). *L. johnsonii* L531 pretreatment reduced *E. coli* adhesion to PMECs by competitive exclusion and the production of inhibitory factors and decreased *E. coli*-induced destruction of cellular morphology and ultrastructure. *E. coli* induced activation of NLRP3 inflammasome associated with increased expression of NLRP3, ASC, and cleaved caspase-1, however, *L. johnsonii* L531 inhibited *E. coli*-induced activation of NLRP3 inflammasome. Up-regulation of interleukin *(Il)-1β*, *Il-6*, *Il-8*, *Il-18*, tumor necrosis factor alpha, and chemokine *Cxcl2* expression after *E. coli* infection was attenuated by *L. johnsonii* L531. *E. coli* infection inhibited autophagy, whereas *L. johnsonii* L531 reversed the inhibitory effect of *E. coli* on autophagy by decreasing the expression of autophagic receptor SQSTM1/p62 and increasing the expression of autophagy-related proteins ATG5, ATG16L1, and light chain 3 protein by Western blotting analysis. Our findings suggest that *L. johnsonii* L531 pretreatment restricts NLRP3 inflammasome activity and induces autophagy through promoting ATG5/ATG16L1-mediated autophagy, thereby protecting against *E. coli*-induced inflammation and cell damage in PMECs.

## 1. Introduction

Coliform Mastitis (CM), one of the most significant cardinal symptoms of postpartum dysgalactia syndrome in sows [1,2,3], poses serious economic implications for the pig farming as both the health of postpartum sows and the growth of their piglets can be severely damaged by mastitis-induced lactation failure [4]. Among responsible influencing factors, *Escherichia coli* is of importance because of its prevalence and the damage this bacterium may induce. The main management strategies for preventing and treating CM involve the overuse and misuse of antibiotics, which goes generally ineffective, but actually increases the risk of spread of antimicrobial resistance to commensal and opportunistic bacteria. Therefore, it is urgent to develop antibiotic alternatives, especially in the veterinary field. 

Probiotics, such as *Lactobacillus,* have anti-inflammatory potential and can attenuate systemic proinflammatory immune responses [5,6,7,8,9]. A recent finding demonstrated that *L. johnsonii* significantly inhibited pathogen infection such as *E. coli*, *S. enteritidis*, and *Staphylococcus aureus* [10]. Our recent study showed that *L. johnsonii* L531, a probiotic bacterium isolated from the intestinal contents of healthy newly weaned piglets, exhibits the ability to reduce pathogen load and maintain metabolic balance [11]. *L. johnsonii* L531 also ameliorates enteritis of newly weaned piglets during *S*. infantis infection [12]. However, whether *L. johnsonii* L531 could provide a new strategy for preventing and improving CM and the exact molecular mechanism underlying probiotic actions remains to be elucidated.

Mammary epithelial cells (MECs) play an important role in initiating and driving an immediate and rapidly innate immune response when pathogens have overcome physiological barriers and entered the mammary through the ducts of the mammary gland [13]. MECs react rapidly to invading microbes by binding to pathogen-associated molecular patterns (PAMP) via pattern recognition receptors (PRRs), such as transmembrane Toll-like receptors (TLR) and oligomerization domain NOD-like receptors (NLR), and inducing the production of pro- and anti-inflammatory cytokines [14,15]. Toll-like receptor 4 (TLR4), binds to ligand lipopolysaccharide (LPS) in Gram-negative bacteria ligand delivery or recognition to activate NF-κB and induce gene transcription encoding the inactive pro-IL-1β precursor [16]. 

Besides, the inflammasome, a multi-protein complex, is of important in the innate immune system. In inflammasome complex platform, the adaptor protein ASC recruits caspase-1 and leads to auto-proteolytic processing of pro-caspase-1 into its active form, which triggers the release of proinflammatory cytokines IL-1β and IL-18 [17,18]. At present, several inflammasomes have been described, such as NLRP3, NLRC4 and AIM2. *Nlrp3*-deficient mice susceptible to experimental colitis and decreased expression of IL-1 as well as the anti-inflammatory cytokine IL-10 [19]. However, excessive activation of the NLRP3 inflammasome results in disorders of autoimmune and metabolic [20]. The NLRP3 inflammasome can be a new target for treating mastitis, avoiding the development of antibiotic resistance [21].

Autophagy is essential for maintaining cell biological homeostasis under pressurized conditions that also play important roles in many innate and adaptive immune processes [22,23,24]. Given that pathogen infections represent one of the major threats to the health of animals and human beings, thus, autophagy pathway is necessary to outcome host-microbe interactions. For example, *E. coli* O157:H7 dampens autophagy via a Tir-dependent pathway mechanism while autophagy activation suppresses the adhesion of *E. coli* O157:H7 to intestinal epithelial cells [25]. Moreover, recent evidence highlighted the cross-talk between autophagy and NLRP3 inflammasome activation [26]. Autophagy deficiency causes the accumulation of damaged mitochondria, resulting in NLRP3-ASC assembly [27]. An obvious elevation of IL-1β and IL-18 expression is observed in macrophages from ATG16L1-deficient mice infected with LPS [28]. Autophagy captures and degrades inflammasomes through protein ubiquitination, resulting in the recruitment of SQSTM1 and LC3 [29]. Our recent study found that *L. rhamnosus* GG dampens autophagy to protect against diarrhea induced by *Salmonella* Infantis and ameliorates intestinal inflammation in piglets [30]. It remains to be determined whether *L. johnsonii* L531 participates in the regulation of autophagy to relieve inflammation in CM. 

In the present study, a PMEC model of *E. coli* infection was established to test the hypothesis that *L. johnsonii* L531 attenuates *E. coli*–induced inflammation and cell damages by restricting NLRP3 inflammasome activity and promoting ATG5/ATG16L1–mediated autophagy in PMECs.

## 2. Materials and Methods

### 2.1. Ethics Statement

All animals were treated in strict accordance with the Guidelines for Laboratory Animal Use and Care from the Chinese Center for Disease Control and Prevention and the Rules for Medical Laboratory Animals (1998) from the Chinese Ministry of Health, under protocol CAU20151001-1, which was approved by the Animal Ethics Committee of the China Agricultural University.

### 2.2. Porcine Mammary Epithelial Cell Culture

PMECs were kindly provided by Prof. Guoyao Wu of Texas A&M University (Texas A&M University, College Station, TX, USA). The PMECs were maintained in Dulbecco’s Modified Eagle Medium/Ham’s F-12 medium (DMEM/F12) supplemented with 10% heat-inactivated fetal bovine serum (Thermo Scientific, Waltham, MA, USA), 5 μg/mL of insulin, 5 ng/mL of epidermal growth factor, 1 μg/mL of hydrocortisone, 50 μg/mL of gentamycin and 1× PSN (penicillin-G, streptomycin, and neomycin) antifungal/antibiotics [31] at 37 °C in a humidified 5% CO_2_ atmosphere. And the cells were trypsinized with a 0.25% trypsin solution containing 1 mM EDTA.

### 2.3. Bacterial Strains and Growth Conditions

*Lactobacillus johnsonii* L531 was obtained from a healthy newly weaned piglet in our laboratory [11]. In our experiments, *L. johnsonii* L531 was inoculated into fresh De Man, Rogosa, and Sharpe (MRS) broth (Oxid, Hampshire, UK) at a ratio of 1:100 and grown for 18 h until reaching the OD600 of 0.5 at 37 °C under microaerophilic conditions. Bacteria were washed three times by centrifugation at 3000× *g* for 10 min at 4 °C, and then resuspended in sterile physiologic saline. *L. johnsonii* L531 was plated on MRS agar after serial dilutions and quantified by determining the amount of colony forming unit (CFU).

*Escherichia coli* O111:K58 (CVCC1450), purchased from China Institute of Veterinary Drug Center (Beijing, China), was grown in Luria-Bertani (LB) broth (Oxoid) at 37 °C with constant shaking to reaching mid-log phase. Recombinant *E. coli* strain constitutively expressing red fluorescent protein (RFP) was constructed using the plasmid pFPV-mCherry with ampicillin resistance by electroporation, cultured in LB broth containing ampicillin, and shaken overnight at 37 °C. Bacteria were then diluted 1:100 in fresh LB broth containing ampicillin (100 μg/mL), and grown for about 2.5 h until reaching the mid-log phase. As mentioned above, mCherry-*E. coli* was washed three times by centrifugation at 3000× *g* for 10 min at 4 °C, resuspended in sterile physiologic saline, and quantified by determination of CFU after serial dilutions.

### 2.4. Immunofluorescence

PMECs (4 × 10^4^ cells/well) were seeded onto 24-well cell culture plates, and cells were divided into three groups: blank control group (cells without mCherry-*E. coli* treatment but treated with primary antibody), negative control group (cells treated with mCherry-*E. coli* and treated with PBS instead of primary antibody) and model group (cells infected with mCherry-*E. coli* and treated with primary antibody). On day 2, cells were washed three times with phosphate-buffered saline (PBS) for 5 min and infected with mCherry-*E. coli* (2 × 10^6^ CFU). At 6 h after mCherry-*E. coli* infection, cells were washed three times with PBS for 5 min to removed non-adherent mCherry-*E. coli*, and then fixed with 4% paraformaldehyde for 15 min on ice. Cells were rinsed three times with PBS for 5 min and incubated for 45 min in blocking solution (1× PBS/5% goat serum/0.3% Triton X-100) to reduce non-specific background. Then PMECs were incubated with mouse anti-cytokeratin-18 antibody (1:200 dilution, ab668, Abcam, Cambridge, UK) for overnight at 4 °C, and were washed three times with PBS, following by secondary antibody FITC-labeled anti-mouse IgG (1:100 dilution, F4143, Sigma-Aldrich, Budapest, Hungary) for 1 h at room temperature. The 4′,6′-diamidino-2-phenylindole (DAPI; Sigma-Aldrich, Budapest, Hungary) was used to stain cell nuclei for 5 min at 37 °C. Cells were visualized and imaged using an Olympus BX41 microscope (Olympus, Tokyo, Japan) equipped with a Canon EOS 550D camera head (Canon, Tokyo, Japan). The experiment was performed three independent times.

### 2.5. Adhesion Assay

The adhesion assay was performed, as previously described [7]. Briefly, PMECs (4 × 10^5^ cells/well) were seeded on a 6-hole cell culture plate. After cultured for 24 h, PMECs were pretreated with (i) DMEM/F12 medium alone, (ii) live *L. johnsonii* L531 (2 × 10^7^ CFU), (iii) UV-irradiated *L. johnsonii* L531 (2 × 10^7^ CFU), (iv) heat-killed *L. johnsonii* L531 (2 × 10^7^ CFU), (v) *L. johnsonii* L531 supernatant (pH 7.2), or (vi) DMEM/F12 medium containing lactic acid (pH 6.2) for 3 h. PMECs were washed three times with phosphate-buffered saline (PBS) and exposed to *E. coli* (2 × 10^7^ CFU). At 6 h after *E. coli* challenge, the monolayer cells were washed four times with PBS to remove non-adherent bacteria and then were harvested by 0.05% trypsin treatment for 10 min at 37 °C. The amount of *E. coli* recovered was cultured on LB agar containing 100 μg/mL ampicillin and quantified by measuring colony-forming unit (CFU), as described above. The inhibition of adhesion was calculated by taking the adhesion of *E. coli* without any treatment as positive controls (100% adhesion) by the formula described. The experiment was performed three independent times.

### 2.6. Internalization Assay

The internalization assays were conducted as previously described [7]. Briefly, cells were treated with *L. johnsonii* L531 (2 × 10^7^ CFU) or *E. coli* (2 × 10^7^ CFU). At 6 h after treatment, the number of internalized *L. johnsonii* L531 or *E. coli* was determined by adding 100 μg/mL of gentamicin to kill extracellular bacteria. The amount of *E. coli* and *L. johnsonii* L531 recovered was cultured on the selective growth plate LB agar containing ampicillin and MRS agar, respectively, and both quantified by measuring colony-forming unit (CFU), as described above. The experiment was performed three independent times.

### 2.7. Scanning Electron Microscopy (SEM) and Transmission Electron Microscopy (TEM)

PMECs were subjected to four conditions: (i) medium alone; (ii) *E. coli* alone (2 × 10^7^ CFU) at a multiplicity of infection (MOI) of 50:1; (iii) incubation with *L. johnsonii* L531 (2 × 10^7^ CFU) for 3 h; or (iv) pre-incubation with *L. johnsonii* L531 (2 × 10^7^ CFU) for 3 h prior to exposure to *E. coli*. Cells were washed three times with PBS after incubation with *L*. *johnsonii* L531 for 3 h before exposure to *E. coli*. At 6 h after *E. coli* infection, followed by four rinses with PBS to remove the non-adherent bacteria, a monolayer of cultured PMECs was harvested and fixed with 3% glutaraldehyde (pH 7.4). The cells were observed using a Quanta 200 FEG scanning electron microscope (FEI, Eindhoven, The Netherlands) and an H7500 transmission electron microscope (Hitachi, Tokyo, Japan), respectively. The experiment was performed three independent times.

### 2.8. Cell Death Assay

Cytotox96 cytotoxicity assay (Promega, Madison, WI, USA) was applied to assess the cell death by detecting LDH levels according to the following formula ((LDH infected − LDH uninfected)/(LDH total lysis − LDH uninfected)) × 100. The experiment was performed three independent times.

### 2.9. Real-Time Quantitative PCR

At 3, 6 and 9 h after *E. coli* infection, total RNA was extracted from PMECs for gene expression analysis using TRIzol reagent (Invitrogen, Carlsbad, CA, USA). An ABI 7500 real-time PCR system (Applied Biosystems, Foster City, CA, USA) was used for quantitative real-time PCR analyses. The sequences of the primers used were listed in Table 1. Relative mRNA expression data were shown as fold-change according to the 2^−ΔΔCT^ method as previously described [15]. Data of gene expression were normalized to the glyceraldehyde-3-phosphate dehydrogenase (*Gapdh*) gene. The experiment was performed three independent times.

### 2.10. Western Blotting

As stated above, PMECs (4 × 10^5^ cells/well) were subjected to the following conditions: (i) medium alone; (ii) *E. coli* alone (2 × 10^7^ CFU); (iii) incubation with *L. johnsonii* L531 (2 × 10^7^ CFU) for 3 h; or (iv) pre-incubation with *L. johnsonii* L531 (2 × 10^7^ CFU) for 3 h prior to exposure to *E. coli* (2 × 10^7^ CFU). Total proteins were extracted from the PMECs for Western blotting assay. The following primary antibodies included rabbit polyclonal anti-NLRP3 (1:2000 dilution, 19771-1-AP), rabbit polyclonal anti-ASC (1:500 dilution, 10500-1-AP), rabbit polyclonal anti-ATG 5 (1:1000 dilution, 10181-1-AP), rabbit polyclonal anti-ATG 16L1 (1:1000 dilution, 19812-1-AP), rabbit polyclonal anti-sequestosome 1 (SQSTM1) (1:500 dilution, 18420-1-AP) (ProteinTech Group, Rosemont, IL, USA), rabbit polyclonal anti-LC3A/B (1:1000 dilution, 12741) (Cell Signaling Technology, Danvers, MA, USA), and rabbit polyclonal anti-caspase-1 (1:1000 dilution, ab179515) (Abcam). To verify equal sample loading, the membrane was incubated with mouse anti-β-actin (1:5000 dilution, 66009-1-Ig), mouse anti-GAPDH (1:5000 dilution, 60004-1-Ig), and rabbit anti-β-tubulin (1:1000 dilution, 10094-1-AP) (ProteinTech Group). HRP-conjugated anti-mouse IgG (1:5000 dilution, SA00001-1) or anti-rabbit IgG (1:5000 dilution, SA00001-2) (ProteinTech Group) were used as secondary antibodies. Band intensity was quantified by densitometric analysis using ImageJ software (National Institutes of Health, Bethesda, MD, USA). The experiment was performed three independent times.

### 2.11. Statistical Analysis

SAS statistical software, version 9.3 (SAS Institute Inc., Cary, NC, USA), was used for statistical analysis. With regard to small sample sizes, normal distribution and homogeneity of variance were assumed using the UNIVARIATE (Shapiro–Wilk test) and HOVTEST procedures. Natural logarithm transformation was performed prior to analysis for *Il-1β* and *Il-18* data to yield a normal distribution. Data of normally distributed adhesion ratio, LDH, real-time PCR and Western blotting were evaluated using analysis of variance procedures. Data are expressed as mean ± SEM of three independent experiments. Differences between means were assessed using Tukey’s honestly significant difference test for post hoc multiple comparisons. A value of *p* < 0.05 was considered statistically significant.

## 3. Results

### 3.1. L. johnsonii L531 Pretreatment Reduces the Adhesion of E. coli to PMECs

To visualize the localization of *E. coli* O111:K58 in PMECs, an *E. coli* strain harboring the pFPV-mCherry plasmid as described in materials and methods was successfully created, and was observed clearly under the inversed fluorescent microscope (Appendix A). Immunofluorescence analysis of fixed PMECs infected with recombinant *E. coli* O111:K58 showed positive staining for cytokeratin-18, a specific marker of the epithelial cell lineage, and adherent RFP-enriched *E. coli* was detected on the surface of PMECs at 6 h after infection (Figure 1A). In the experiment, the insertion of a recombinant plasmid expressing RFP had no effect on the growth activity of *E. coli* O111:K58 (Appendix A). Besides, high genetic stability of pFPV-mCherry plasmid was observed across time, which indicated that recombinant *E. coli* O111:K58 can be used in in vitro experiments or animal infection studies regardless of the long experimental period.

After 6 h following *E. coli* stimulation, the number of adherent *E. coli* was 1.48 × 10^6^ ± 1.74 × 10^4^ CFU (means ± SEM). Pretreatment with live, UV-irradiated and culture supernatants of *L. johnsonii* L531 result in the reduction of *E. coli* adhesion rate to 27.22% (*p* < 0.001), 65.16% (*p* = 0.003), and 73.86% (*p* = 0.025), respectively, whereas no change was observed in cells pretreated with heat-killed *L. johnsonii* L531 and DMEM/F12 medium acidified with lactic acid (Figure 1B). The internalization of *E. coli* by PMECs was not observed. Compared with *E. coli* infection only, there was no alteration for the count of *E. coli* in the supernatant regardless of treatment (Figure 1C). 

### 3.2. L. johnsonii L531 Pretreatment Reduces PMEC Damage Induced by E. coli

The morphology of monolayer cells was observed under the general light microscope to determine whether *L. johnsonii* L531 makes any difference to PMECs infected with *E. coli* (Appendix A). The untreated control PMECs presented a cobblestone-like appearance as well as a conservative density of cell proliferation. However, following infection with *E. coli*, the PMEC monolayer was severely disrupted and the cells showed partly vacuolation. *L. johnsonii* L531 pretreatment effectively reduced *E. coli*-induced cell damage up to 6 h after *E. coli* infection.

### 3.3. L. johnsonii L531 Pretreatment Ameliorates Disruption of PMEC Ultrastructure and Decreased Cell Death Induced by E. coli

Under SEM, the untreated control PMECs appeared plump, with intact cell membranes and abundant slender microvilli on the surface of cells. However, at 6 h after *E. coli* infection, cell membranes were disrupted, microvilli became less evident and *E. coli* could be seen attached to the surface of cells. PMECs incubated with *L. johnsonii* L531 alone remained cell membrane integrity, and exhibited evident microvilli. Pretreatment with *L. johnsonii* L531 decreased the degree of cell membrane disruption for up to 6 h after *E. coli* infection, and both *E. coli* and *L. johnsonii* L531 could be seen attached to the surface of PMECs. *E. coli* is straight and short rod-shaped, which is thicker than *L. johnsonii* L531, while *L. johnsonii* L531 is elongated and rod-shaped, which is thinner and longer than *E. coli* (Figure 2A).

Under conventional TEM (Figure 2B), in untreated control PMECs, organelles in the cytoplasm were structurally complete and distinct exhibited homogeneous electron density, normal mitochondrial structures, and integrity endoplasmic reticulum. In contrast, *E. coli* infection caused abruption of microvilli, loosening of the cytoplasmic matrix structure, disorganization, large vesicular nuclei, and hazy mitochondria. PMECs incubated with *L. johnsonii* L531 alone maintained a normal appearance. Pretreatment with *L. johnsonii* L531 alleviated the damage of PMEC ultrastructure for up to 6 h after *E. coli* infection, and mitochondria were readily visible within autophagosomes with a characteristic double-membrane structure.

Cell death was assessed by measuring the release of LDH at 6 h after *E. coli* infection (Figure 2C). Obviously, *E. coli* infection increased the percentage of dead cells (*p* < 0.001), while *L. johnsonii* L531 pre-incubation effectively decreased the *E. coli*–induced PMEC death.

### 3.4. L. johnsonii L531 Pretreatment Ameliorates E. coli-Induced Activation of NLRP3 Inflammasome

There were no significant changes in *Tlr4* mRNA expression of PMECs at 3 h after *E. coli* infection or *L. johnsonii* L531 pretreatment. Compared with untreated control cells, the mRNA expression of *Tlr4* was increased in cells infected with *E. coli* alone at 6 and 9 h after challenge (*p* < 0.001 and *p* < 0.001, respectively; Figure 3A), whereas *L. johnsonii* L531 pretreatment prior to *E. coli* infection led to a decrease in *Tlr4* mRNA expression in PMECs in comparision to cells only infected with *E. coli* at 6 and 9 h (*p* = 0.002 and *p* = 0.003, respectively). Moreover, no differences in *Tlr4* mRNA expression were observed between control cells and cells pretreated with *L. johnsonii* L531 alone at 6 h post-infection.

Upon *E. coli* challenge, the *Nlrp3* mRNA expression was upregulated at 3, 6 and 9 h post-infection in PMECs compared with untreated control cells (*p* < 0.001), but this increase was attenuated by *L. johnsonii* L531 pretreatment (*p* < 0.001, *p* < 0.001, and *p* = 0.038, respectively; Figure 3B). The Western blotting analysis was consistent with the qRT-PCR results, at 6 h, the PMECs exposed to *E. coli* alone had higher NLRP3 protein expression than the untreated control cells and cells only pretreated with *L. johnsonii* L531 (*p* < 0.001 and *p* < 0.001), while *L. johnsonii* L531 pretreatment followed by *E. coli* infection resulted in a decrease in NLRP3 protein expression in comparison to the cells only exposed to *E. coli* (*p* < 0.001, Figure 3C). Likewise, expression of ASC protein was elevated at 6 h after *E. coli* infection compared with the untreated control cells and cells only pretreated with *L. johnsonii* L531 (*p* < 0.001 and *p =* 0.002), whereas this up-regulation was inhibited by pre-incubation with *L. johnsonii* L531 (*p* = 0.013, Figure 3D). Pretreatment with *L. johnsonii* L531 also attenuated *E. coli*-induced increase in the expression of active isoform caspase-1 p10 protein (*p* = 0.006, Figure 3E).

### 3.5. L. johnsonii L531 Pretreatment Suppresses E. coli-Induced Cytokine and Chemokine mRNA Expression in PMECs

With a particular focus on differentially expressed genes associated with inflammatory response, qRT-PCR was conducted to investigate the defense-related biological pathways of PMECs to *E. coli* infection.

Compared with untreated control PMECs, an immediate and strong up-regulation of *Il-1β* mRNA expression in cells infected with *E. coli* only was observed at all time points tested (*p* < 0.001), but not in cells incubated with *L. johnsonii* L531 only (Figure 4A). The increase in *Il-1β* mRNA expression induced by *E. coli* was attenuated by pre-incubation with *L. johnsonii* L531 at all time points (*p* = 0.001, *p* = 0.012, and *p* < 0.001, respectively). Similar profile of changes was observed in *Tnf-α* and *Cxcl2* mRNA expression during *E. coli* infection (Figure 4E,F).

There was a dramatic up-regulation in the level of *Il-18* expression in cells with *E. coli* infection alone, at different time points tested (3, 6, and 9 h), when compared with that in untreated control PMECs (*p* = 0.004, *p* < 0.001, and *p* < 0.001, respectively; Figure 4B). In PMECs, preincubate with *L. johnsonii* L531 led to a considerable decrease in the expression of *Il-18* in comparision to cells only infected with *E. coli* at 6 h and 9 h (*p* < 0.001, and *p* < 0.001, respectively). 

The increase of *Il-6* mRNA expression in cells infected with *E. coli* alone was statistically significant when compared with control PMECs at various time points after *E. coli* challenge (*p* < 0.001, *p* < 0.001, and *p* = 0.012, respectively). Whereas pretreatment with *L. johnsonii* L531 attenuated *E. coli*–induced increase in *Il-6* expression at 3 h and 6 h (*p* < 0.001, and *p* < 0.001, respectively; Figure 4C) 

Likewise, an early increase in *Il-8* mRNA expression was observed upon infection with *E. coli* at 3 h and 6 h post-infection (*p* < 0.001, and *p* < 0.001, respectively). Pretreatment with *L. johnsonii* L531 decreased the level of *Il-8* expression in comparision to cells infected with *E. coli* alone at 3 and 6 h (*p* < 0.001, and *p =* 0.031, respectively; Figure 4D), whereas this phenomenon was not observed at later time points.

### 3.6. L. johnsonii L531 Pretreatment Reverses the Inhibitory Effect of E. coli on Autophagy in PMECs

Compared with untreated control cells, the expression of ATG5 protein was decreased at 6 h after *E. coli* infection in cells infected with *E. coli* alone (*p* = 0.026), while there was a statistically dramatic increase in ATG5 protein expression in cells incubated with *L. johnsonii* L531 alone or pretreated with *L. johnsonii* L531 followed by *E. coli* infection when compared with that in cells only infected with *E. coli* (*p* = 0.005 and *p* = 0.001, Figure 5A). Similarly, a lower expression of ATG16L1 protein was found in cells only infected with *E. coli* in comparision to the cells of untreated control (*p* = 0.019, Figure 5B). Compared with the cells infected with *E. coli* alone, an elevated expression of ATG16L1 protein in cells incubated with *L. johnsonii* L531 alone or pretreated with *L. johnsonii* L531 followed by *E. coli* infection was observed (*p* < 0.001, and *p* < 0.001), even than in cells of untreated control (*p* < 0.001 and *p* < 0.001). Interestingly, at 6 h, PMECs infected with *E. coli* alone had higher expression of SQSTM1 protein than did the cells of untreated control (*p* = 0.047, Figure 5C). However, SQSTM1 protein expression was decreased in PMECs incubated with *L. johnsonii* L531 or the group of *L. johnsonii* L531 pretreatment followed by *E. coli* infection than in PMECs infected with *E. coli* alone (*P* < 0.001, and *P* = 0.001), even lower than in untreated control cells (*p* = 0.001, and *p* = 0.023). There was a down-regulation in the expression of LC3A/B-II protein in PMECs infected with *E. coli* alone than in untreated contro cells (*p* = 0.049), but *L. johnsonii* L531 pretreatment upregulated LC3A/B-II protein expression compared with cells infected with *E. coli* alone (*p* = 0.010; Figure 5D). PMECs incubated with *L. johnsonii* L531 alone had higher expression of LC3A/B-II protein than did cells of untreated control, cells infected with *E. coli* alone or pretreated with *L. johnsonii* L531 (*p* = 0.001, *p* < 0.001, and *p* = 0.005, respectively).

To further explore the effect of *L. johnsonii* L531 on autophagy in PMECs induced by *E. coli*, cells were treated with Chloroquine (CQ) to inhibit autophagic degradation. Cells were treated with 20 μM CQ for 2 h (optimum working conditions shown in Figure 5E) and then the expression of LC3A/B-II was detected. Western blot results showed that cells treated with *L. johnsonii* L531 had higher expression of LC3A/B-II protein than did the control cells with or without CQ treatment (*p* = 0.001, and *p* = 0.020, respectively; Figure 5E). As expected, CQ increased the LC3A/B-II accumulation in both treated cells (*p* = 0.036 and *p* = 0.037). Compared with cells that CQ treatment alone, cells co-incubated with *L. johnsonii* L531 and CQ had higher expression of LC3A/B-II protein (*p* = 0.001), indicating *L. johnsonii* L531 still increased the LC3A/B-II accumulation when autophagic degradation was inhibited.

Furthermore, the autophagy inhibitor 3-methyladenine (3-MA) was used to inhibit the synthesis of autophagosomes. Cells were treated with 2.5 mM 3-MA for 12 h (optimum working conditions shown in Figure 5F) and then the expression of LC3A/B-II and SQSTM1 was detected. As shown in Figure 5F, in the absence of 3MA, compared with the untreated control cells, the LC3A/B-II expression was obviously increased (*p* = 0.003), but the SQSTM1 expression was obviously decreased (*p* = 0.310) in cells treated with *L. johnsonii* L531. Moreover, 3MA reduced the expression of LC3A/B-II (*p* = 0.003 and *p* = 0.039) but elevated the expression of SQSTM1 (*p* = 0.001 and *p* = 0.047) in both control cells and cells treated with *L. johnsonii* L531. Notably, in cells co-incubated with *L. johnsonii* L531 and 3-MA, LC3A/B-II expression was increased compared with cells treated with 3-MA alone (*p* = 0.035), and SQSTM1 expression was decreased in cells co-incubated with *L. johnsonii* L531 and 3-MA than cells treated with 3-MA alone (*p* = 0.001).

## 4. Discussion

Probiotics are widely used in the treatment of mastitis for their good antibacterial activity and multiple biological functions. Oral application of *Lactobacillus salivarius* PS2 in adequate amounts during late pregnancy can be an efficient approach to prevent infectious mastitis for women [32]. Some studies have shown that *lactic acid bacteria* (LAB) used in animal feed at the dry-off period constitutes an alternative tool for bovine mastitis prevention [33]. *L. rhamnosus* GR-1 is considered as a beneficial microbe to reduce bovine mastitis via the activities of inhibiting pathogen adhesion and immunomodulatory capacity [7]. However, Lactobacilli have species/strain-specific characteristics in response to external factors such as defense against pathogen infections. A meta-analysis showed that the effectiveness of preventing strategies in pathogen infection using probiotic supplements is associated with species of specific probiotic [34]. *L. johnsonii* L531 has been revealed to improve nutrient digestion and absorption, inhibit pathogeny microbiology growth and promote microbiota homeostasis of the intestinal tract in piglets during the critical weaning period [11]. Additionally, our recent study showed that administration of *L. johnsonii* L531 ameliorates enteritis of newly weaned piglets during *S*. infantis infection by promoting IgA secretion, reducing inflammation, and eliminating the damaged mitochondria [12]. In the present study, we found that pretreatment with *L. johnsonii* L531 restraints *E. coli* adhesion due to competitive exclusion and the production of inhibitory factors, thereby ameliorating *E. coli*–induced cell damages. *L. johnsonii* L531 also attenuates *E. coli*–induced inflammation by restricting NLRP3 inflammasome activity and promoting ATG5/ATG16L1-mediated autophagy in PMECs.

The adhesion to host epithelial cells is a prerequisite for *E. coli* successfully infection [35]. In this research, preincubation of PMECs with live, UV-irradiated and culture supernatants of *L. johnsonii* L531 had no killing effect on *E. coli* directly but did result in an obvious reduction of cell-adherent *E. coli* compared with that observed in PMECs infected with *E. coli*. The necessity of pretreatment implies that there is a competition for binding sites. Surface layer proteins and exopolysaccharides of probiotics are likely to make a significant contribution to establishing prior colonization and exclusion of pathogens by repressing the formation of pathogenic biofilms via inhibiting adhesion [36,37]. The competitive exclusion of pathogens from the surface of PMECs by *L. johnsonii* L531 proposed to be one of the potential mechanisms for inhibiting *E. coli* infections. Moreover, some metabolites produced by probiotics, such as bacteriocins, indole, and H_2_O_2_, can inhibit the growth and colonization by pathogenic bacteria [36,38]. Although the bacterium-specific mechanism by which *L. johnsonii* L531 exerted the benefits remains unclear, our data suggest that in addition to directly suppress *E. coli* adhesion by competitive exclusion or the production of inhibitory factors, *L. johnsonii* L531 can indirectly affect or interfere with *E. coli* infections by stimulating host immune defenses.

The innate immune system, being the first line of defense against pathogen infection, which helps initiate long-lasting adaptive immunity to provide enhanced protection against subsequent reinfection by the same microorganism. Innate recognition of PAMP is mediated by evolutionary conserved PRR such as membrane-associated TLRs and cytoplasmic NOD-like receptors, mostly stimulated by NF-ĸB signaling [39]. For instance, TLR2 identifies cell wall components of gram-positive bacteria [40,41], whereas TLR4 identifies LPS from gram-negative bacteria [16,42]. Studies show that TLR4 is the main pattern recognition receptor in response to *E. coli*-induced mastitis in mice, goats and dairy cows [43,44,45]. We found that *E. coli*–induced an up-regulation of *Tlr4* in PMECs at 6 h after *E. coli* infection. Activation of TLR4-dependent signaling may recruit inflammatory cells and promote the process of mastitis during *E. coli* infection. 

TLRs promotes the activation of the NLRP3 inflammasome, which is the key sensor of cell or tissue damage. With an N-terminal pyrin domain, NLRP3 interacts with the inflammasome adaptor protein ASC through interactions between pyrin domains. ASC also has a caspase recruitment domain, which recruits caspase-1 via self-oligomerization, to contribute to the activation of the protease, caspase-1 [19,46]. And then the downstream pathway is activated, thereby promoting the expression of NF-κB and the secretion of inflammatory cytokines, such as IL-1β and IL-18, which play a key role in recruiting neutrophils to sites of infection and promote inflammatory anti-bacterial immune responses [18,47]. We found that pre-incubation with *L. johnsonii* L531 restricts *E. coli*-induced NLRP3 inflammasome activation, including the expression of receptor protein NLRP3, the adaptor protein ASC, the effector caspase-1 p10 and the release of *Il-1β* and *Il-18*. Consistent with other studies in vivo and the microarray analysis results [48,49,50,51], the expression of cytokines and chemokines, such as *Il-6*, *Il-8*, *Tnf-α*, and *Cxcl2* was up-regulated after *E. coli* challenge, whereas this increase was attenuated by pre-incubation with *L. johnsonii* L531. *L. johnsonii* L531 inhibits the activation of NLRP3 inflammasome during *E. coli* infection.

In addition to regulating inflammatory responses to protect against pathogen infection, mammary epithelial cells can also utilize autophagy as a protective function against pathogens. Autophagy is an essential protective mechanism of innate immunity for degrading intracellular pathogens. A growing body of evidence suggested that ATG proteins have important roles in the process of autophagy. ATG5 is indispensable for the formation of the autophagic vesicle. Knocking down or knocking out *Atg5* can result in down-regulation or total inhibition of autophagy. ATG5-ATG12-ATG16 complex is an ubiquitin-like complex that is required for autophagosome formation [52]. Knockout of *Atg5* or *Atg16l1* in intestinal epithelial cells leads to diminished resistance to *Salmonella* Typhimurium [53,54]. It was also reported that probiotic *Bacillus amyloliquefaciens* induces autophagy to promote the elimination of intracellular *E. coli* in macrophages via upregulating the expression of Beclin1 and ATG5-ATG12-ATG16L1 complex, but not Akt/mTOR signaling pathway [55]. LC3 is conjugated with PE via a ubiquitin-like system and converted to LC3-II with the help of ATG4, ATG7, and ATG3 [56]. We showed that *L. johnsonii* L531 pretreatment increased the expression of ATG5, ATG16L1, as well as LC3A/B-II. In addition, when using CQ to inhibit autophagic degradation, we found that *L. johnsonii* L531 still increased the LC3A/B-II accumulation. Our findings suggest that *L. johnsonii* L531 promotes the synthesis of autophagosome during *E. coli* infection.

Alternatively, SQSTM1/p62, a well-known substrate of selective autophagy, can directly interact with LC3 on the isolation membrane and then be recruited into the autophagosome and degraded. Autophagy defect causes SQSTM1 aberrantly accumulation, leading to the formation of large aggregates including SQSTM1 and ubiquitin [26]. Impaired selective turnover of SQSTM1 is a primary cause of liver injury in autophagy-deficient mice [57]. The accumulation of SQSTM1 is indicative of aborted autophagy flux. In the present study, *E. coli* infection inhibited autophagy by decreasing the expression of LC3A/B-II, and increasing the expression of autophagic receptor SQSTM1/p62. We found that *L. johnsonii* L531 pretreatment markedly elevated the expression of LC3A/B-II protein, and decreased the expression of SQSTM1/p62 protein, indicating that *L. johnsonii* L531 could effectively improve the autophagy level of PMECs may due to its effect on promoting autophagic degradation. In addition, after 3-MA was used to inhibit the synthesis of autophagosomes, LC3A/B-II expression increased in cells co-incubated with *L. johnsonii* L531 and 3-MA compared with cells treated with 3-MA alone, and SQSTM1 expression was decreased in cells co-incubated with *L. johnsonii* L531 and 3-MA than the untreated control cells. Our data indicates that *L. johnsonii* L531 could promote autophagic degradation during *E. coli* infection. 

Autophagy negatively regulates the activation of NLRP3 inflammasome through various mechanisms [24,52,58], including directly inhibition of NLRP3 inflammasome activation by eliminating sources of endogenous NLRP3 inflammasome agonists, such as damaged mitochondria and mitochondrial DNA [27,59], suppression of IL-1β secretion by targeting pro-IL-1β for lysosomal degradation [60], and selective degradation of inflammasome components, such as NLRP3 and ASC [29,61]. Thus, autophagy deficiency leads to the accumulation of damaged mitochondria and the release of mitochondrial DNA, causing enhanced ROS generation and NLRP3-ASC assembly [62]. It was reported that macrophage-specific deletion of autophagy genes in mice leads to inflammasome-mediated IL-1β release and uveitis, which is an inflammation-mediated eye disease often observed in patients with CD [63]. In this study, *E. coli* infection resulted in autophagy inhibition in PMECs, which might cause enhanced NLRP3-ASC assembly and ROS generation. Whereas pretreatment with *L. johnsonii* L531 inhibited the *E. coli*–induced activation of NLRP3 inflammation, and promoted ATG5/ATG16L1-mediated autophagy, thereby ameliorating *E. coli*-induced inflammation and cell damage in PMECs. Alternatively, *L. johnsonii* L531 might stimulate autophagy by promoting the synthesis of autophagosomes, thus partially reducing the activation of NLRP3 inflammasome.

The application of *L. johnsonii* L531 to prevent *E. coli* infection in PMECs leads to a promising perspective on new strategies to improve the efficiency of mastitis treatment. This PMEC model provides an in vitro framework for evaluating response to *L. johnsonii* L531-based intervention in porcine mastitis. However, the involvement of autophagy in regulating NLRP3 inflammasome activation in PMECs after *E. coli* challenge and whether NLRP3 inflammasome activation is reduced by *L. johnsonii* L531-mediated autophagy is still undefined. Remarkably, no cell model can ever be a perfect analog of animals themselves. Undoubtedly, the primary result of this study requires more measurement of inflammasome activity and autophagosome function to explore the cross-talk between NLRP3 inflammasome activation and autophagy. And future in vivo studies including the optimal dosage of *L. johnsonii* L531 in clinical application and further research on the bacterium-specific mechanism will be essential for realizing the potential of *L. johnsonii* L531 strain against mastitis.

## 5. Conclusions

In conclusion, in this study, *E. coli* infection resulted in excessive activation of NLRP3 inflammasome and inhibited autophagy, while pretreatment with *L. johnsonii* L531 ameliorates *E. coli*–induced inflammation and cell damage in PMEC model. On the one hand, *L. johnsonii* L531 directly suppresses the adhesion of *E. coli* by competitive exclusion and the production of inhibitory factors thereby decreasing the *E. coli* stimulation in PMECs and subsequently alleviating *E. coli*–induced inflammation and cell damage. On the other hand, once *E. coli* adheres to PMECs, *L. johnsonii* L531 pretreatment restricts NLRP3 inflammasome activity and weakens the inhibitory effect of *E. coli* on autophagy through promoting ATG5/ATG16L1-mediated autophagy, which, in turn, ameliorating the *E. coli*-induced inflammation in PMECs (Figure 6). Our results suggest that *L. johnsonii* L531 attenuates *E. coli*-induced inflammation and cell damage through restricting NLRP3 inflammasome activity and promoting ATG5/ATG16L1-mediated autophagy in PMECs.

## Figures and Tables

**Figure 1 vetsci-07-00112-f001:**
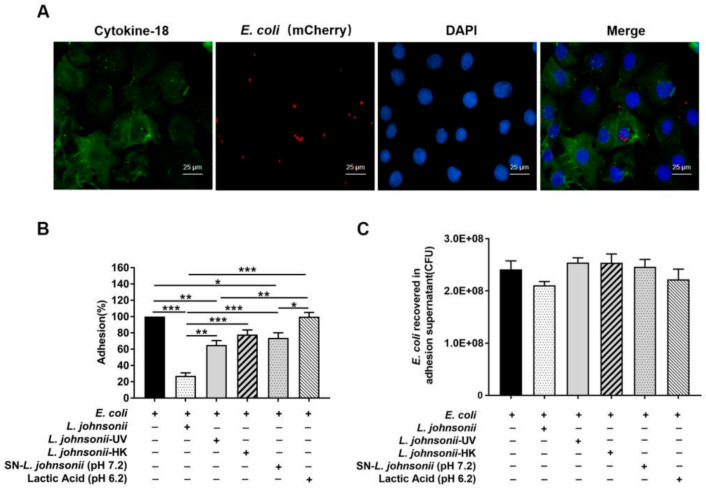
Pretreatment with *L. johnsonii* L531 decreased the adhesion of *E. coli* to PMECs. (**A**) Immunofluorescence analysis of the *E. coli* adhesion (using mCherry-*E. coli*) to the primary cultured PMECs. Representative confocal immunofluorescence images showing typical morphology of pure PMECs, which were stained with mouse anti-cytokeratin-18 (green) and DAPI (blue). Scale bar, 25 μm. Data were representative of three separate experiments. (**B**) The adhesion rate of *E. coli* in different treatments. PMECs were harvested at 6 h after *E. coli* infection. The adhesion assay using *E. coli* alone serves as a reference. The adhesion rate was presented as the ratio of the number of adhered *E. coli* in different treatments to the reference number of adhered *E. coli*. (**C**) The amount of *E. coli* recovered in the supernatant of adhesion experiments. Upon *E. coli* challenge, the amount of *E. coli* recovered was counted in the supernatant of adhesion experiments at 6 h post-infection. Data were exhibited as the mean ± SEM from three separate experiments. * *p* < 0.05, ** *p* < 0.01, *** *p* < 0.001.

**Figure 2 vetsci-07-00112-f002:**
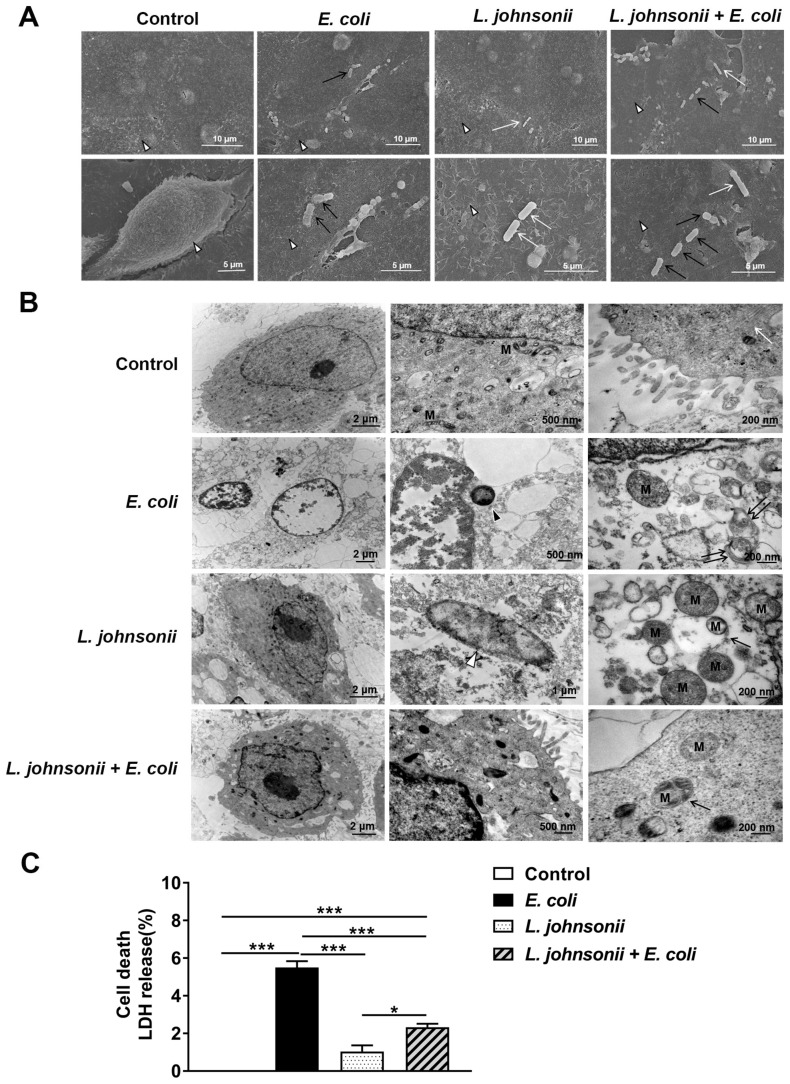
Pre-incubation with *L. johnsonii* L531 reduced *E. coli*-induced cell death and disruption of PMEC ultrastructure. PMECs were treated with medium alone (Control), mCherry–*E. coli* (*E. coli*), *L. johnsonii* L531 alone (*L. johnsonii*), or they were preincubated with *L. johnsonii* L531 for 3 h followed by *E. coli* challenge (*L. johnsonii* + *E. coli*). PMECs ultrastructure observed using SEM (**A**) and TEM (**B**). The subcellular structure of PMECs harvested from the cultured PMECs at 6 h after *E. coli* infection. In (**A**), white arrowheads indicate microvilli on the surface of PMECs, black arrows indicate *E. coli* and white arrows indicate *L. johnsonii* L531. In (**B**), black arrows indicate autophagosome with enclosing double membrane, M indicates mitochondria, white arrows indicate endoplasmic reticulum structures, black arrowheads indicate *E. coli*, white arrowheads indicate *L. johnsonii* L531, and double black arrows indicate marrow shaped structures. (**C**) LDH was applied to assay the PMECs death by detecting LDH levels in supernatants, which were collected from the indicated cells at 6 h after *E. coli* infection. Data were shown as the mean ± SEM of three independent experiments. * *p* < 0.05, *** *p* < 0.001.

**Figure 3 vetsci-07-00112-f003:**
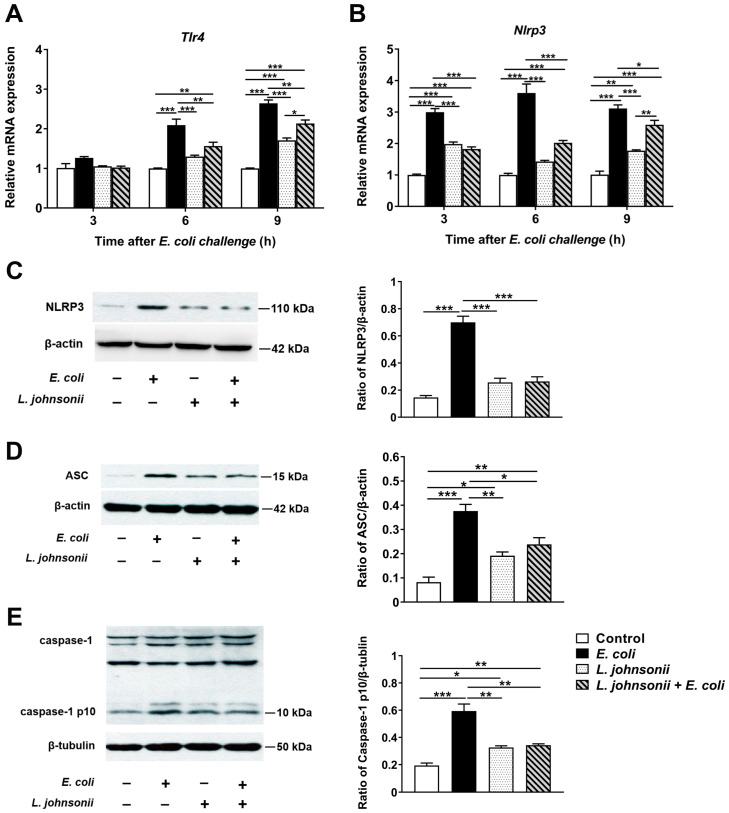
Pretreatment with *L. johnsonii* L531 inhibited *E. coli*-induced NLRP3 inflammasome activation. The mRNA expression of *T**lr4* (**A**) and *Nlrp3* (**B**) genes obtained from the indicated PMECs at 3, 6, 9 h post-infection was analyzed by quantitative real-time PCR. Western blotting analysis of NLRP3 (**C**), ASC (**D**), and caspase-1 (**E**) expression in PMECs at 6 h after challenge with *E. coli* or cells pre-incubated with *L. johnsonii* L531 for 3 h before *E. coli* infection. Data were presented as the mean ± SEM of three independent experiments. * *p* < 0.05; ** *p* < 0.01; *** *p* < 0.001.

**Figure 4 vetsci-07-00112-f004:**
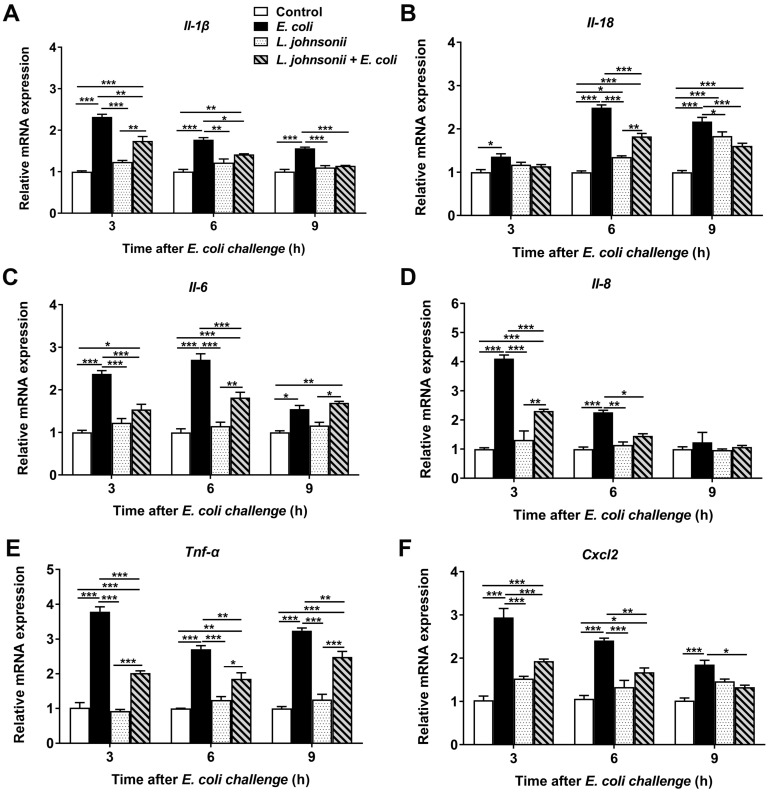
The effect of *L. johnsonii* L531 on cytokine and chemokine mRNA expression in PMECs infected with *E. coli*. The mRNA expression of *Il-1β* (**A**), *Il-18* (**B**), *Il-6* (**C**), *Il-8* (**D**), *Tnf-α* (**E**), and *Cxcl2* (**F**) in cells obtained from the indicated PMECs at 3, 6, 9 h post-infection was analyzed by quantitative real-time PCR. Data were presented as the mean ± SEM of three independent experiments. * *p* < 0.05, ** *p* < 0.01, *** *p* < 0.001.

**Figure 5 vetsci-07-00112-f005:**
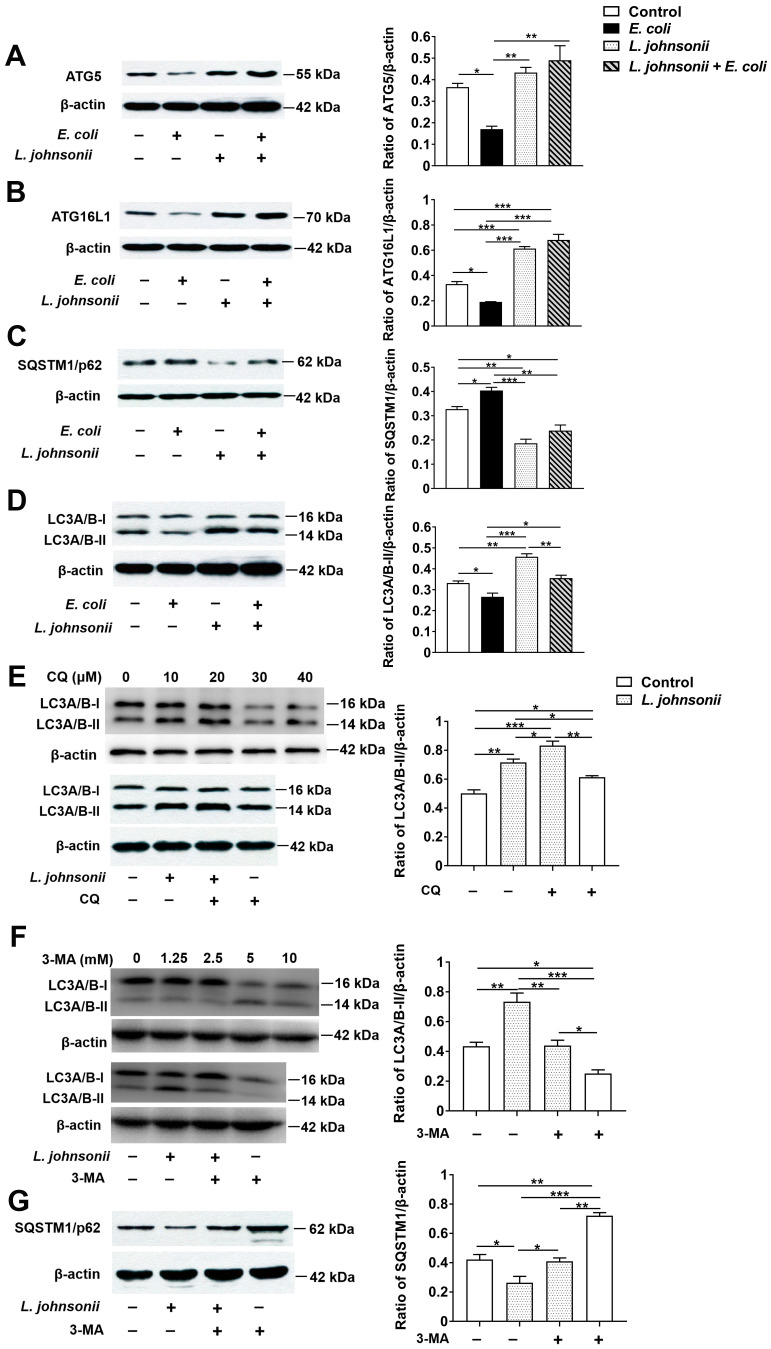
Pre-incubation with *L. johnsonii* L531 inhibited *E. coli*-induced autophagy in PMECs. Western blotting analysis of (**A**) ATG5, (**B**) ATG16L1, (**C**) SQSTM1, (**D**) LC3A/B in PMECs collected at 6 h post-infection (left panels). Representative panels showing expression of ATG5, ATG16L1, SQSTM1, and LC3A/B. Results are exhibited as the ratio of protein band intensity for ATG5, ATG16L1, SQSTM1, and LC3A/B-II to the intensity of the β-actin band (right panels), (**E**) PMECs treated with the different working conditions of CQ for 2 h to inhibit autophagic degradation. The addition of optimum working conditions of CQ led to the prominent accumulation of LC3A/B-II in both different groups. PMECs treated with the different working conditions of 3-MA for 12 h to inhibit the synthesis of autophagosomes. Western blotting analysis of (**F**) LC3A/B-II and (**G**) SQSTM1 in cells treated with optimum working conditions of 3MA. Data were presented as the mean ± SEM of three independent experiments. * *p* < 0.05; ** *p* < 0.01; *** *p* < 0.001.

**Figure 6 vetsci-07-00112-f006:**
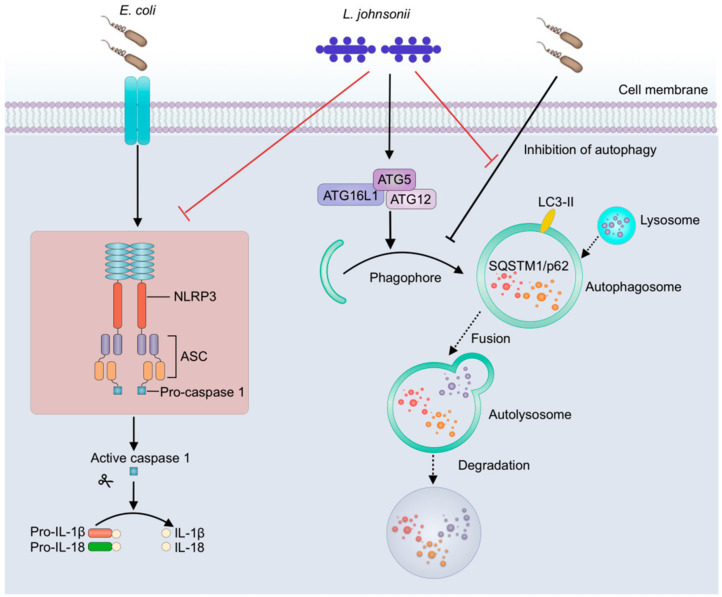
*L. johnsonii* L531 restricts NLRP3 inflammasome activity and weakens the inhibitory effect of *E. coli* on autophagy through promoting ATG5/ATG16L1-mediated autophagy. *L. johnsonii* L531 attenuates *E. coli*-induced inflammation and cell damage through restricting NLRP3 inflammasome activity and promoting ATG5/ATG16L1-mediated autophagy in PMECs. Full lines represent the results of the present study, and dashed lines represent the conclusions drawn in other studies.

**Table 1 vetsci-07-00112-t001:** Real-time PCR primers.

Primers Name	Direction ^a^	Sequence (5′→3′)	Accession Number
*Gapdh*	F	CCAGAACATCATCCCTGCTT	NM_001206359
	R	GTCCTCAGTGTAGCCCAGGA	
*Il-1β*	F	GGCCGCCAAGATATAACTGA	NM_214055
	R	GGACCTCTGGGTATGGCTTTC	
*Il-18*	F	GCTGCTGAACCGGAAGACAA	NM_213997.1
	R	AAACACGGCTTGATGTCCCT	
*Il-6*	F	GGGAAATGTCGAGGCTGTG	NM_214399
	R	AGGGGTGGTGGCTTTGTCT	
*Il-8*	F	TCCTGCTTTCTGCAGCTCTC	NM_213867
	R	GGGTGGAAAGGTGTGGAATG	
*Tnf-α*	F	GCCCACGTTGTAGCCAATGTCAAA	NM_214022
	R	GTTGTCTTTCAGCTTCACGCCGTT	
*Cxcl2*	F	GGAAGTTTGTCTCAACCCCGC	NM_001001861
	R	AGCCAGTAAGTTTCCTCCATCTC	
*Tlr4*	F	GCCATCGCTGCTAACATCATC	NM_001113039
	R	CTCATACTCAAAGATACACCATCGG	
*Nlrp3*	F	GAGCCTAGGAACTCGGAGGA	NM_001256770.1
	R	GCTCATCAAAGGCACCTTGC	

^a^ F = forward; R = reverse.

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
