# Peer review of "Lactobacillus johnsonii L531 Ameliorates Escherichia coli-Induced Cell Damage via Inhibiting NLRP3 Inflammasome Activity and Promoting ATG5/ATG16L1-Mediated Autophagy in Porcine Mammary Epithelial Cells"

_vetsci, 2020, doi:10.3390/vetsci7030112_

Round 1

Reviewer 1 Report

The authors examined the effects of L. Johnsonii L531 on attenuating E. coli-induced inflammatory damage in porcine mammary epithelial cells. Authors found that L. Johnsonii L531 inhibited E. coli-induced activation of NLRP3 inflammasome and attenuated E. coli-induced IL family and TNF-alpha expression. Moreover, L. Johnsonii L531 promoted autophagy. Therefore, authors concluded that L. Johnsonii L531 restricts NLRP3 inflammasome and induces autophagy, thereby protecting against  activity and induces pretreatment restricts NLRP3 inflammasome activity and induces autophagy through promoting ATG5/ATG16L1-mediated autophagy, thereby protecting against E. coli-induced inflammation and cell damage.

The manuscript contains interesting results that merit publication, but I have several comments to improve the manuscript.

Major concerns

  1. Figure 5-E, F

Based on the result of western blot for SQSTM1, I think that E. coli- blocks the degradation of phagosome. I am interested in the SQSTM1 expression levels when cell treated with CQ or 3-MA treatment.

  1. How Johnsonii L531 promote the autophagy ? Authors need to discuss about this point.

Minor

Figure 3, Figure 5 : It is not clear what each bar represents.

Line218-221: So what?

Author Response

Veterinary Sciences

The Editor

06 August 2020                                         Re: vetsci-860270

Dear Editor

Please find enclosed a revised version of our manuscript entitled “Lactobacillus johnsonii L531 ameliorates Escherichia coli–induced cell damage via inhibiting NLRP3 inflammasome activity and promoting ATG5/ATG16L1-mediated autophagy in porcine mammary epithelial cells” (vetsci-860270). The manuscript has been revised according to the comments of the editor and the reviewers and is resubmitted for your consideration. We have used editing service to polish the English language of our manuscript.

We would like to thank the reviewers and editors of our manuscript for the interest they have expressed in our study and for their constructive comments. We have extensively revised the manuscript and use track changes in the revision to answer their criticisms and comments. Please find below a point by point response to the raised concerns. We hope that this manuscript is now suitable for publication in Veterinary Science.

Sincerely,

Prof. Dr. Jiufeng Wang

College of Veterinary Medicine

China Agricultural University

Tel: 86 10 6273 1094

Fax: 86 10 6273 1274

Reviewer #1

Comments and Suggestions for Authors

The authors examined the effects of L. johnsonii L531 on attenuating E. coli-induced inflammatory damage in porcine mammary epithelial cells. Authors found that L. johnsonii L531 inhibited E. coli-induced activation of NLRP3 inflammasome and attenuated E. coli-induced IL family and TNF-alpha expression. Moreover, L. Johnsonii L531 promoted autophagy. Therefore, authors concluded that L. johnsonii L531 restricts NLRP3 inflammasome and induces autophagy, thereby protecting against activity and induces pretreatment restricts NLRP3 inflammasome activity and induces autophagy through promoting ATG5/ATG16L1-mediated autophagy, thereby protecting against E. coli-induced inflammation and cell damage.

The manuscript contains interesting results that merit publication, but I have several comments to improve the manuscript.

Major concerns

  1. Figure 5-E, F

Based on the result of western blot for SQSTM1, I think that E. coli- blocks the degradation of phagosome. I am interested in the SQSTM1 expression levels when cell treated with CQ or 3-MA treatment.

Anwer: We acknowledge the reviewer’s concern. In the revised manuscript, we have supplemented the experiment of SQSTM1 expression level after PMECs were treated with 3-MA and the Figure 5G has been updated.

The result is presented as “Cells were treated with 2.5 mM 3-MA for 12 h (optimum working conditions shown in Figure 5F) and then the expression of LC3A/B-II and SQSTM1 was detected...and SQSTM1 expression was decreased in cells co-incubated with L. johnsonii L531 and 3-MA than cells treated with 3-MA alone (P = 0.001). (lines 433-444).

The discussion is shown as “In addition, after 3-MA was used to inhibit the synthesis of autophagosomes. LC3A/B-II expression increased in cells co-incubated with L. johnsonii L531 and 3-MA compared with cells treated with 3-MA alone, and SQSTM1 expression was decreased in cells co-incubated with L. johnsonii L531 and 3-MA than the untreated control cells. Our data indicates that L. johnsonii L531 could promote autophagic degradation during E. coli infection.” (lines 555-560).

  1. How johnsonii L531 promote the autophagy? Authors need to discuss about this point.

Answer: We acknowledge the reviewer’s concern and have carefully re-discussed this point in the revised manuscript.

 “ATG5-ATG12-ATG16 complex is an ubiquitin-like complex that is required for autophagosome formation [52]. LC3 is conjugated with PE via a ubiquitin-like system and converted to LC3-II with the help of ATG4, ATG7, and ATG3 [56]. We showed that L. johnsonii L531 pretreatment increased the expression of ATG5, ATG16L1, as well as LC3A/B-II. In addition, when using CQ to inhibit autophagic degradation, we found that L. johnsonii L531 still increased the LC3A/B-II accumulation. Our findings suggest that L. johnsonii L531 promotes the synthesis of autophagosome during E. coli infection.”

“Alternatively, SQSTM1/p62, a well-known substrate of selective autophagy, can directly interact with LC3 on the isolation membrane and then be recruited into the autophagosome and degraded. The accumulation of SQSTM1 is indicative of aborted autophagy flux. In the present study, E. coli infection inhibited autophagy by decreasing the expression of LC3A/B-II, and increasing the expression of autophagic receptor SQSTM1/p62. We found that L. johnsonii L531 pretreatment markedly elevated the expression of LC3A/B-II protein, and decreased the expression of SQSTM1/p62 protein, indicating that L. johnsonii L531 could effectively improve the autophagy level of PMECs may due to its effect on promoting autophagic degradation. In addition, after 3-MA was used to inhibit the synthesis of autophagosomes, LC3A/B-II expression increased in cells co-incubated with L. johnsonii L531 and 3-MA compared with cells treated with 3-MA alone, and SQSTM1 expression was decreased in cells co-incubated with L. johnsonii L531 and 3-MA than the untreated control cells. Our data indicates that L. johnsonii L531 could promote autophagic degradation during E. coli infection.” (lines 529-558) 

Minor

  1. Figure 3, Figure 5 : It is not clear what each bar represents.

Answer: We acknowledge the reviewer’s concern. We have labeled each bar in Figure 3 and Figure 5.

  1. Line218-221: So what?

Answer: We acknowledge the reviewer’s concern. We have carefully rewritten the sentence in the revised manuscript (lines 239-240).

Reviewer 2 Report

Zou and collaborators studied, in vitro, the use of the probiotic Lactobacillus Johnsonii L531 to prevent Escherichia coli-induced mammary tissue damage. They demonstrated that a pretreatment by L. Johnsonii reduced E. coli adhesion and E. coli-induced cell damage, inflamasome activation and autophagy.

Comments

For western blot, the full picture need to be added as a supplementary data with a square indicating which portion is presented in the manuscript. The western blot protein ladder should appear on each figure with the molecular weight.

The methods need to be developed and more details are needed. For instance, bacteria have been cultured in hypoxic condition? How the E. coli growth has been quantified, which technics? Time, controls, concentration of the secondary for the immunochemistry? Experiments have been reproduced three time but how many wells have been used per experiment? …. Authors need to check carefully the method and to give more details.

The N is too low for a ANOVA analysis; authors need to use a non –parametric test such as a Kruskal Wallis test followed by Dunn multiple comparison posthoc.

When using “as previously described”, (page 5 for instance) a reference is needed.

Paragraph 3.2, authors should temper their conclusions as there is no quantification, it is just an observation.

Paragraph 3.3, the figure supporting the observation described in the two first sentences should be indicated. The dot after “less evident. and” should be remove.

Paragraph 3.4, “Moreover, no differences in TLR4 mRNA expression were observed between control cells and cells pretreated with L. johnsonii L531 followed by E. coli infection at 6 h.”  Figure 3 there is a significant (**) between control and L. johnsonii L531 + E. coli at 6h which do not support the sentence.

Authors should check carefully the nomenclature. Genes (mRAN expression results) should be in italic and lower case. Sometimes authors wrote Johnsonnii, sometimes johnsonnii.

Author Response

Veterinary Sciences

The Editor

06 August 2020                                         Re: vetsci-860270

Dear Editor

Please find enclosed a revised version of our manuscript entitled “Lactobacillus johnsonii L531 ameliorates Escherichia coli–induced cell damage via inhibiting NLRP3 inflammasome activity and promoting ATG5/ATG16L1-mediated autophagy in porcine mammary epithelial cells” (vetsci-860270). The manuscript has been revised according to the comments of the editor and the reviewers and is resubmitted for your consideration. We have used editing service to polish the English language of our manuscript.

We would like to thank the reviewers and editors of our manuscript for the interest they have expressed in our study and for their constructive comments. We have extensively revised the manuscript and use track changes in the revision to answer their criticisms and comments. Please find below a point by point response to the raised concerns. We hope that this manuscript is now suitable for publication in Veterinary Science.

Sincerely,

Prof. Dr. Jiufeng Wang

College of Veterinary Medicine

China Agricultural University

Tel: 86 10 6273 1094

Fax: 86 10 6273 1274

Reviewer #2

Zou and collaborators studied, in vitro, the use of the probiotic Lactobacillus johnsonii L531 to prevent Escherichia coli-induced mammary tissue damage. They demonstrated that a pretreatment by L. johnsonii reduced E. coli adhesion and E. coli-induced cell damage, inflamasome activation and autophagy.

Comments

  1. For western blot, the full picture need to be added as a supplementary data with a square indicating which portion is presented in the manuscript. The western blot protein ladder should appear on each figure with the molecular weight.

Answer: We acknowledge the reviewer’s concern. Here we provide original representative image as follows:

  1. The methods need to be developed and more details are needed. For instance, bacteria have been cultured in hypoxic condition? How the coligrowth has been quantified, which technics? Time, controls, concentration of the secondary for the immunochemistry? Experiments have been reproduced three time but how many wells have been used per experiment? …. Authors need to check carefully the method and to give more details.

Answer: We acknowledge the reviewer’s concern. In the revision, we have added more details in materials and methods (lines 128-243).

(1) The growth condition of L. johnsonii L531 is presented as “L. johnsonii L531 was inoculated into fresh De Man, Rogosa, and Sharpe (MRS) broth (Oxid, Hampshire, UK) at a ratio of 1:100 and grown for 18 h until reaching the OD600 of 0.5 at 37°C under microaerophilic conditions.” (lines 127-129).

(2) E. coli was quantified by determination of CFU after serial dilutions, which is shown as “As mentioned above, mCherry-E. coli was washed three times by centrifugation at 3000 × g for 10 min at 4℃, resuspended in sterile physiologic saline, and quantified by determination of CFU after serial dilutions.” (lines 140-142).

(3) The immunochemistry is presented as “PMECs (4 × 104 cells/well) were seeded onto 24-well cell culture plates, and cells were divided into three groups: blank control group (cells without mCherry-E. coli treatment but treated with primary antibody), negative control group (cells treated with mCherry-E. coli and treated with PBS instead of primary antibody) and model group (cells infected with mCherry-E. coli and treated with primary antibody)”(lines 144-148).

“At 6 h after mCherry-E. coli infection, cells were washed three times with PBS for 5 min to removed non-adherent mCherry-E. coli” (lines 150-151).

The concentration of the secondary antibody is presented as “secondary antibody FITC-labeled anti-mouse IgG (1:100 dilution, F4143, Sigma-Aldrich)” in lines 156-157.

(4) All experiments in our study were performed three independent times, and there were three duplicates in each group for per experiment. For instance, in the experiment of Western blotting (lines 214-217), PMECs were subjected to the following conditions: (i) medium alone (Control); (ii) E. coli alone (E. coli); (iii) incubation with L. johnsonii L531 (L. johnsonii) for 3 h; or (iv) pre-incubation with L. johnsonii L531 for 3 h prior to exposure to E. coli (L. johnsonii + E. coli). The control group including Control1, Control2 and Control3 three parallel repetitions, the E. coli group including E. coli1, E. coli2 and E. coli3 three parallel repetitions, as an analogy, 12 wells of PMECs have been used per experiment. Likewise, in the experiment of Immunofluorescence, PMECs “were divided into three groups, including blank control group (cells treated with mCherry-E. coli and treated with PBS instead of primary antibody) and model group (cells infected with mCherry-E. coli and treated with primary antibody).”(lines 144-148), 9 wells of PMECs have been used per experiment.

  1. The N is too low for a ANOVA analysis; authors need to use a non –parametric test such as a Kruskal Wallis test followed by Dunn multiple comparison posthoc.

Answer: We thank for the reviewer’s kind concern. Our experiment was performed three independent times, and there were three parallel repetitions in each group. Data of adhesion ratio, LDH, real-time PCR and Western blotting follow the law of normal distribution, which were evaluated using analysis of variance procedures. Data are expressed as mean ± SEM of three independent experiments. Differences between means were assessed using Tukey’s honestly significant difference test for post hoc multiple comparisons. A value of P < 0.05 was considered statistically significant. (lines 238-243).

  1. When using “as previously described”, (page 5 for instance) a reference is needed.

Answer: We acknowledge the reviewer’s concern. We have carefully modified the sentence “as described in materials and methods” in the revised manuscript in line 247.

5.Paragraph 3.2, authors should temper their conclusions as there is no quantification, it is just an observation.

Answer: We agree with the comments and thanks for the reviewer’s sound advice. We have carefully rewritten the conclusion in paragraph 3.2 presented as: “L. johnsonii L531 pretreatment reduces PMEC damage induced by E. coli” in line 279.

6.Paragraph 3.3, the figure supporting the observation described in the two first sentences should be indicated. The dot after “less evident. and” should be remove.

Answer: We acknowledge the reviewer’s concern. A new improved image of control group was presented and marked in Figure 2A in the revised manuscript. “Under SEM, the untreated control PMECs appeared plump, with intact cell membranes and abundant slender microvilli on the surface of the cells. However, at 6 h after E. coli infection, cell membranes were disrupted, microvilli became less evident and E. coli could be seen attached to the surface of cells.” (lines 290-293). The legend of Figure 2 is shown as “white arrowheads indicate microvilli on the surface of PMECs” (lines 319-320). And the dot after “less evident. and” has been removed (line 292).

7.Paragraph 3.4, “Moreover, no differences in TLR4 mRNA expression were observed between control cells and cells pretreated with L. johnsonii L531 followed by E. coli infection at 6 h.” Figure 3 there is a significant (**) between control and L. johnsonii L531 + E. coli at 6h which do not support the sentence.

Answer: We acknowledge the reviewer’s concern. We have carefully rewritten the paragraph 3.4 based on our results: “Compared with untreated control cells, the mRNA expression of Tlr4 was increased in cells infected with E. coli alone at 6 and 9 h after challenge (P < 0.001 and P < 0.001, respectively; Figure 3A),whereas L. johnsonii L531 pretreatment prior to E. coli infection led to a decrease in Tlr4 mRNA expression in PMECs in comparision to cells only infected with E. coli at 6 and 9 h (P = 0.002 and P = 0.003, respectively). Moreover, no differences in Tlr4 mRNA expression were observed between control cells and cells pretreated with L. johnsonii L531 alone at 6 h post-infection.” (lines 331-338).

8.Authors should check carefully the nomenclature. Genes (mRAN expression results) should be in italic and lower case. Sometimes authors wrote Johnsonnii, sometimes johnsonnii.

Answer: We acknowledge the reviewer’s concern and thanks for the reviewer’s sound advice. We have carefully checked the nomenclature in the full-text, and genes in mRAN expression results have been rendered in italic and lower case. The “johnsonii” was used in the revised manuscript.

Round 2

Reviewer 1 Report

Thank you for your complete work.